# Absence of warmth permits epigenetic memory of winter in *Arabidopsis*

Jo Hepworth [1], Rea L. Antoniou-Kourounioti [1], Rebecca H. Bloomer[1], Catja Selga [2], Kristina Berggren [3], Deborah Cox[1], Barley R. Collier Harris[1], Judith A. Irwin [1], Svante Holm [3], Torbjörn Säll[2], Martin Howard [1] & Caroline Dean[1]

Plants integrate widely fluctuating temperatures to monitor seasonal progression. Here, we investigate the temperature signals in field conditions that result in vernalisation, the mechanism by which flowering is aligned with spring. We find that multiple, distinct aspects of the temperature profile contribute to vernalisation. In autumn, transient cold temperatures promote transcriptional shutdown of *Arabidopsis FLOWERING LOCUS C* (*FLC*), independently of factors conferring epigenetic memory. As winter continues, expression of VERNALIZA-TION INSENSITIVE3 (VIN3), a factor needed for epigenetic silencing, is upregulated by at least two independent thermosensory processes. One integrates long-term cold temperatures, while the other requires the absence of daily temperatures above 15 °C. The lack of spikes of high temperature, not just prolonged cold, is thus the major driver for vernalisation. Monitoring of peak daily temperature is an effective mechanism to judge seasonal progression, but is likely to have deleterious consequences for vernalisation as the climate becomes more variable.

[1] John Innes Centre, Norwich Research Park, Norwich NR4 7UH, UK. [2] Department of Biology, Lund University, Sölvegatan 35, Lund 223 62, Sweden. [3] Faculty of Science, Technology and Media, Department of Natural Sciences, Mid Sweden University, Sundsvall SE-851 70, Sweden. Jo Hepworth and Rea L. Antoniou-Kourounioti contributed equally. Correspondence and requests for materials should be addressed to M.H. (email: martin.howard@jic.ac.uk) or to C.D. (email: caroline.dean@jic.ac.uk)

Seasonal cues are central to the timing of many developmental transitions in biology. Plants use prolonged cold exposure to align the transition to flowering with spring, a process called vernalisation. In *Arabidopsis*, vernalisation involves the epigenetic silencing of *FLOWERING LOCUS C* (*FLC*) by the conserved Polycomb Repressive Complex 2 (PRC2) in combination with PHD proteins, including cold-induced VERNALIZATION INSENSITIVE3 (VIN3)[1–5]. Vernalisation can occur at constant temperatures between 0 and 15 °C[6–8], but plants in the field experience daily fluctuations of more than 10 °C, equivalent in some locations (e.g. Norwich, UK) to the difference in average temperature between autumn and winter. Experiments indicate that plants are able to extract a reliable signal from this noisy and variable temperature profile[8–11], but how they integrate these signals over timescales of many weeks to judge the passing of winter has not been clear[10,12,13]. Previous analyses have developed cumulative measures of cold exposure over time, including photothermal units or 'degree days', used to plan the harvest dates of crops[9,14–17]. However, the molecular mechanisms underlying these measures are unknown.

Here, we test expression of *FLC* and *VIN3* in the field and in controlled experiments to find that there are two phases of *FLC* downregulation with distinct temperature sensitivities, one independent and one dependent on VIN3. *VIN3* itself is controlled by at least two thermosensors and the circadian clock, which combine to only permit expression when temperatures cease to rise daily above 15 °C. The absence of warmth is therefore a key signal of winter for *Arabidopsis*.

## Results

### *FLC* silencing can be separated into two phases in the field.
To understand the process of vernalisation in natural conditions, we established field experiments (Fig. 1a) in very different winter conditions: a site in Norwich (natural conditions but under glass), and fields in South Sweden and North Sweden, with sowings two weeks apart for the latter to defend against loss to early frost (Supplementary Fig. 1a–c). We monitored the temperature and the expression of *FLC* and *VIN3* to measure the progress of vernalisation in the standard vernalisation-requiring *Arabidopsis thaliana* genotype Col FRI[SF2] (Fig. 1b, c).

Daily ground level temperatures fluctuated widely in Sweden in the early autumn, but the range of fluctuations narrowed in late autumn and early winter as vernalisation occurred, with the temperature at times becoming constant in late winter due to insulating snow cover. In Norwich, the daily temperature variation was also large in the autumn, behaviour which persisted for longer than in Sweden. At all sites *FLC* transcriptional downregulation was detected within the first 40 days. In both Swedish experiments, the *FLC* expression was well described by an exponential decay over time (Fig. 1c), with expression in South Sweden showing a faster rate of decay than in the North (slope significantly different as tested by Analysis of Covariance (ANCOVA), $p = 5.30 \times 10^{-9}$) (Fig. 1d). The plants from the two sowing groups in the North showed very similar *FLC* profiles (Supplementary Fig. 1a–c). *VIN3* showed significant upregulation in the first few weeks of sampling in all Swedish sites, albeit more slowly in the first North experiment than the South (Fig. 1c, Supplementary Figs. 1b and 2a, b). In Norwich, however, the initial downregulation of *FLC* was slow, but later became more rapid than in Sweden ($p = 1.01 \times 10^{-9}$ for slope comparison between Norwich after 55 days and South Sweden) (Fig. 1d), with *VIN3* expression not significantly upregulated until nearly two months after sowing (Fig. 1c). The data for unspliced *FLC* are more variable (Supplementary Fig. 2c), but the faster rate of shutdown in Norwich compared to Sweden is still apparent,

suggesting a reduction in transcription rather than an increase in mRNA degradation rate. It is known that vernalisation efficiency decreases at temperatures close to 0 °C, which may explain the slower rate during the very low temperature period in Sweden[6,8,9].

We hypothesised that the changed *VIN3* behaviour in Norwich, as compared to both Swedish sites, could be the cause of the differences in the *FLC* transcriptional shutdown. We found that *FLC* downregulation in Norwich could be split into two exponential decay regimes with different characteristic rates: the first with a shallower gradient before *VIN3* upregulation, the second with a steeper gradient during *VIN3* upregulation (Fig. 1e, Supplementary Fig. 1d). In a *vin3-4 FRI* null mutant, and mutants of the PRC2 protein VERNALIZATION 2 (VRN2)[2,4,5], the second, faster phase was absent and the slower first phase continued for the whole winter, supporting a causative role for *VIN3* induction and PRC2 in the faster rate (Fig. 1e, Supplementary Fig. 1e). This reduced rate of silencing led to higher *FLC* levels in these mutants, resulting in later flowering at the end of the winter (Supplementary Fig. 3a).

These experiments demonstrated that the VIN3-independent and the VIN3-PRC2-dependent aspects of vernalisation are, in principle, temperature separable phases. Initial transcriptional shutdown is thought to be a requirement for heritable, PRC2-driven epigenetic silencing[18–20]. However, at *FLC*, in constant-condition laboratory experiments and the Swedish field, *VIN3* upregulation occurs simultaneously with VIN3-independent transcriptional shutdown, so the two phases are superimposed (except in PHD-PRC2 mutants)[21,22]. In Norwich, the temperature profile temporally separates these phases, indicating that they are driven by different thermosensory mechanisms.

### *FLC* responds to temperature extremes.
To understand the temperature dependence of both the PRC2-independent and -dependent phases, we investigated the effects of four distinct controlled temperature profiles on *VIN3* and *FLC*. Two of the profiles maintained constant temperatures (8 °C and 14.2 °C), with vernalisation occurring at different rates[7,8]. The other two were fluctuating temperature regimes mimicking profiles from the field experiments, with average temperatures matching those of the constant temperature treatments (Fig. 2a).

The rates of increase of *VIN3* expression in Col FRI[SF2] were very similar in both the constant and fluctuating 8 °C conditions (Fig. 2b). A similar conclusion held for the rates of decrease of *FLC* expression (Fig. 2c). These results support the hypothesis that the two vernalisation phases are similarly responsive to temperatures between 5.5 and 11.5 °C. Indeed, in the *vin3–4 FRI* mutant (Fig. 2d), *FLC* downregulation was again very similar in response to both treatments, suggesting that the VIN3-independent phase does not distinguish between these temperature profiles.

As previously reported[7,8], at constant 14.2 °C as compared to constant 8 °C, both *VIN3* and *FLC* took longer to reach the same level of activation or repression, respectively (Fig. 2b,c). *VIN3* was also upregulated more slowly in response to the 14-fluctuating conditions than the 14-constant (Fig. 2b). The average temperature in the two treatments was the same, so the average temperature is not therefore the signal setting *VIN3* expression. Furthermore, in the 14-fluctuating profile, temperatures as low as 8 °C were experienced. These lower temperatures, when given as a constant treatment, induced *VIN3* more than a constant 14.2 °C (Fig. 2b and ref. [7]). However, the overall effect of a 14-fluctuating profile versus a 14-constant gave much lower *VIN3* expression, suggesting that the high temperature spikes (of up to 22 °C experienced during the 14-fluctuating cycle) are able to prevent

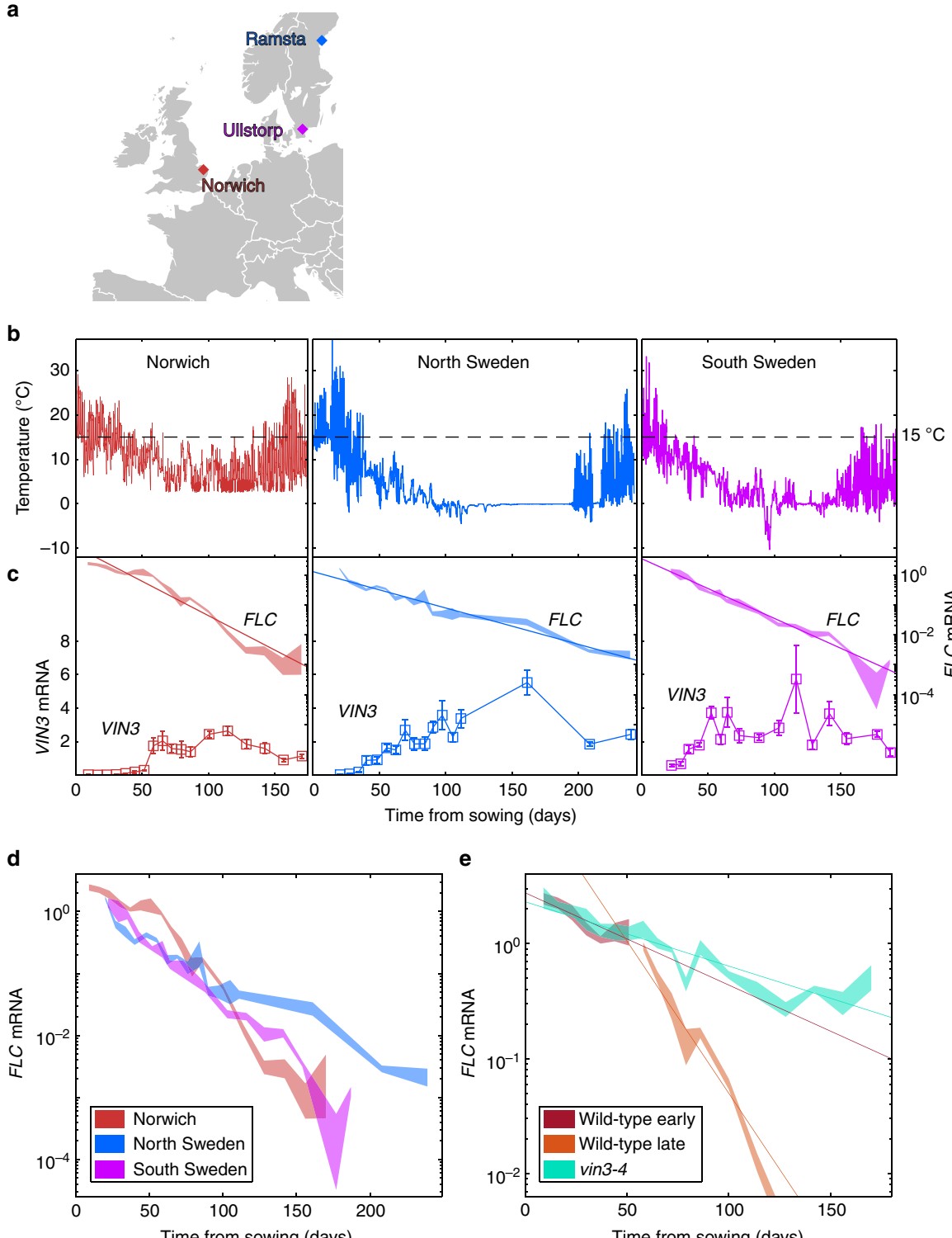

**Fig. 1** *FLC* downregulation occurs in the field in both VIN3-independent and dependent manners. **a** Locations of field sites in Norwich, North Sweden (Ramsta) and South Sweden (Ullstorp). **b** Temperatures experienced by the plants in the different sites. Dashed line indicates 15 °C. In Norwich, glasshouse confinement is probably responsible for buffering low temperatures (see Methods). **c** *FLC* and *VIN3* expression for Col *FRI$^{SF2}$*, as measured in the field corresponding to plot above. *FLC* (top): Thick lines show measured *FLC* mRNA where **c**–**e** thickness of the line represents s.e.m. *VIN3* (below): measured expression of *VIN3* mRNA, errors = s.e.m. **d** Comparison of *FLC* downregulation between different sites. **e** *FLC* downregulation in Norwich for 'wild-type' (Col *FRI$^{SF2}$*) and a mutant impaired in *VIN3* expression (*vin3-4 FRI*). For the wild-type, the data are separated according to presence (late timepoints) or absence (early timepoints) of *VIN3* expression, as measured in panel **c**. RNA levels in **c**–**e** normalised to *UBC*, *PP2A*, and internal control. Straight lines (**c**, **e**) show the best fit exponential decay profile, fitted by least-squares. *n* = 2–6, average ≥ 5. Map adapted from https://commons.wikimedia.org/wiki/File: Blank_map_of_Europe.svg by maix, under CC BY-SA 2.5 license

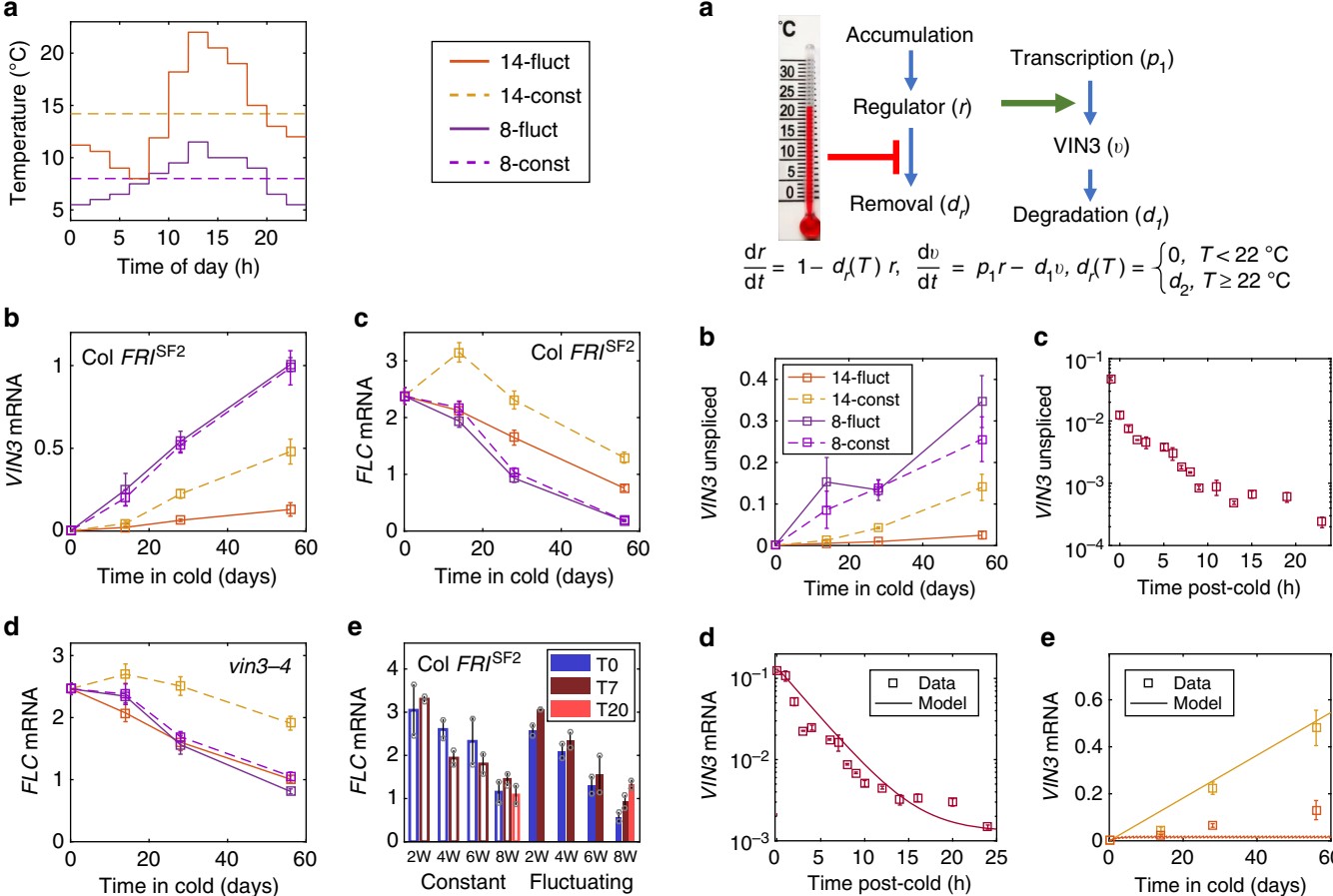

**Fig. 2** *FLC* and *VIN3* expression respond to temperature extremes rather than averages. **a** Temperature profiles of treatments used in the experiment. **b**, **c** *VIN3* and *FLC* mRNA from Col *FRI*^SF2 plants treated with 'cold' temperature conditions of panel **a**. For *VIN3*, plants were sampled 5–7 h after dawn. **b** n = 3–6, average > 5.3. **c** n = 16–21, average > 19. **d** *FLC* mRNA in the mutant *vin3–4*, otherwise as for **c**. n = 9–21, average > 17. **e** *FLC* mRNA in Col *FRI*^SF2 during (T0) and after removal (T7, T20) from the 14.2 °C constant (14-const) or fluctuating (14-fluct) conditions shown in **a**, Tx = x days after return to warm, W = weeks in cold. n = 2 shown as grey circles. Error bars = s.e.m. In **b**–**e** RNA levels were normalised to *UBC*, *PP2A*.

**Fig. 3** Modelling *VIN3* dynamics reveals its temperature inputs. **a** Minimal model of *VIN3* temperature sensing assuming a single thermosensor *r* whose levels quantitatively regulate *VIN3* transcription. $p_1 = 0.91$ day$^{-1}$, $d_1 = 100$ day$^{-1}$, $d_2 = 7$ day$^{-1}$. See Methods for more details. **b** Slow *VIN3* upregulation in the cold for unspliced RNA, sampled 5–7 h after dawn. n = 3–6, average > 4.7. **c**, **d** *VIN3* unspliced and spliced RNA levels upon return to warm (22 °C) following 4 weeks at 5 °C. At time 0, plants were sampled in the cold and subsequently transferred to warm. n = 2–3, average > 2.6. **e** Slow *VIN3* mRNA upregulation in the cold, sampled 5–7 h after dawn. Colours correspond to vernalisation treatment as shown in legend to **b**. n = 3–6, average = 5. Model results shown in **d**, **e**. In **b**–**e**, RNA levels were normalised to *UBC*, *PP2A*; error bars = s.e.m.

high levels of *VIN3* expression. The absence of these warm temperatures thus allows *VIN3* to be expressed at higher levels. This phenomenon could also be seen in the Norwich and North Sweden field experiments (Fig. 1b, c, Supplementary Fig. 2a, b), where *VIN3* expression was low until daily temperature maxima were regularly lower than ~15 °C.

Surprisingly, *FLC* expression reached lower levels more rapidly in the 14-fluctuating versus 14-constant conditions (Fig. 2c), despite the former having lower *VIN3* expression. This result suggested that the greater degree of downregulation of *FLC* in the 14-fluctuating conditions is due to the VIN3-independent shutdown. The *vin3-4 FRI* mutant also showed that VIN3-independent downregulation is as effective in 14-fluctuating as it is at either constant or fluctuating 8 °C, whereas 14-constant is much less effective (Fig. 2d). The simplest interpretation of these results is that the presence of colder temperatures increases the effectiveness of VIN3-independent downregulation and that the lower temperatures in the 14-fluctuating experiment are sufficient to trigger this response. This conclusion may explain results reported by Burghardt et al. in which high-*FLC* genotypes flowered more rapidly when kept in similar fluctuating

conditions[23]. Indeed, in the *vin3–4* mutant, 14-fluctuating conditions can lower *FLC* expression sufficiently to allow more rapid flowering than after 14-constant (Supplementary Fig. 3b).

When plants are moved to warm conditions after weeks of cold temperatures, *FLC* remains silenced[24]. We observed this response in plants treated at 14-constant. However, for plants treated with 14-fluctuating conditions, *FLC* levels reactivated (p = 0.0064 in a one-tailed paired *t*-test), indicating that at least part of the VIN3-independent pathway is not epigenetically stable (Fig. 2e). That reactivation is not always complete is potentially due to non-vernalisation silencing that occurs slowly in the absence of transcription at *FLC*[18].

**VIN3 is controlled by more than one thermosensor.** To understand these results and their implications for temperature sensing in vernalisation, we constructed a simple mathematical model of *VIN3* dynamics to allow us to test and potentially reject hypotheses (Fig. 3a and Methods section). We reasoned that long-term temperature sensing is most likely to act on *VIN3*

transcription initiation because we observed that both spliced (Fig. 2b) and unspliced (Fig. 3b) *VIN3* levels increase slowly over long time periods in the cold. We therefore assumed that *VIN3* transcription is regulated by upstream activating thermosensory processes, which we model with the variable $r$ (Fig. 3a). $r$ slowly responds to time in vernalising temperatures, holds the memory of cold during cold exposure and generates the well-established slow rate of *VIN3* induction[4,25–27]. This model makes minimal assumptions as to the molecular properties of $r$, so our conclusions are applicable to many underlying mechanisms, including cis-based chromatin control, potentially at the *VIN3* locus itself[27,28].

Focusing on the behaviour of *VIN3* around 14.2 °C (fluctuating and constant) as compared to 22 °C, we next asked whether thermosensory behaviour could be entirely isolated within $r$ or whether other thermosensory inputs are required. Two key features must be captured by the model: the slow increase of *VIN3* expression around 14.2 °C (albeit slower at 14-fluctuating than at 14-constant) (Fig. 2b), and a rapid decrease in the warm at 22 °C following a cold treatment (Fig. 3c, d)[4,25–27,29]. The predicted dynamics of $r$ from the model are shown in Supplementary Fig. 4a, b. The degradation of *VIN3* must be fast to show the rapid decrease we see in the warm, and therefore *VIN3* dynamics closely follow those of $r$ (Supplementary Fig. 4c, d). Incorporating both types of thermosensory regulation (cold accumulation, warm removal) into $r$ generated a model that could successfully capture *VIN3* behaviour at 14-constant cold or post-cold 22 °C warm, but not under 14-fluctuating conditions (Fig. 3d, e). In this latter case, $r$ is reset every day at high temperatures and therefore *VIN3* is never expressed at higher levels, in contradiction to our experiments (Fig. 3e). The model therefore demonstrates that the thermosensory process(es) that lead to the fast decrease in *VIN3* in the warm cannot apply to $r$ but must function separately, allowing us to reject a single-thermosensor ($r$) model for *VIN3* regulation.

With this interpretation, there must be at least two thermosensory inputs to *VIN3*. One, $B$, has the slow characteristics of $r$ in the cold and forms a long-term monitor of cold temperatures, similar to degree days (Methods). This input is coupled with a second, separate input ($A$) that monitors current temperatures and is responsible for the rapid decrease in *VIN3* levels at high temperatures, giving the critical absence of warm response for *VIN3*. Since both spliced and unspliced *VIN3* levels decay quickly in the warm (Fig. 3c, d), it is likely that this rapid temperature response is again mediated through control of *VIN3* transcription.

**VIN3 is regulated by the circadian clock.** To further investigate these fast timescale dynamics, we analysed *FLC* and *VIN3* expression during a single day (over 12 h and also over 24 h, Fig. 4a, b, Supplementary Figs. 5 and 6), using both the constant and fluctuating profiles (8 and 14.2 °C average). *FLC* mRNA and unspliced transcript levels did not show an expression pattern consistent with a rapid response (within hours) to changes in temperature, as the pattern was similar in both constant and fluctuating conditions (Supplementary Figs. 5b–g and 6b–i). Furthermore, although *FLC* expression was variable, we did not observe any clear diurnal pattern. In contrast, for *VIN3* mRNA and unspliced levels, we found a strong diurnal pattern in all cases peaking in the afternoon (Figs. 4a, b, Supplementary Figs. 5h–m and 6j–q). As we observed similar patterns for *VIN3* in both constant and fluctuating conditions, this pattern cannot be due to a rapid response to changing temperatures (i.e. it is not explained by the $A$ component). The peak of *VIN3* expression was higher with longer cold treatments, as we would expect from the

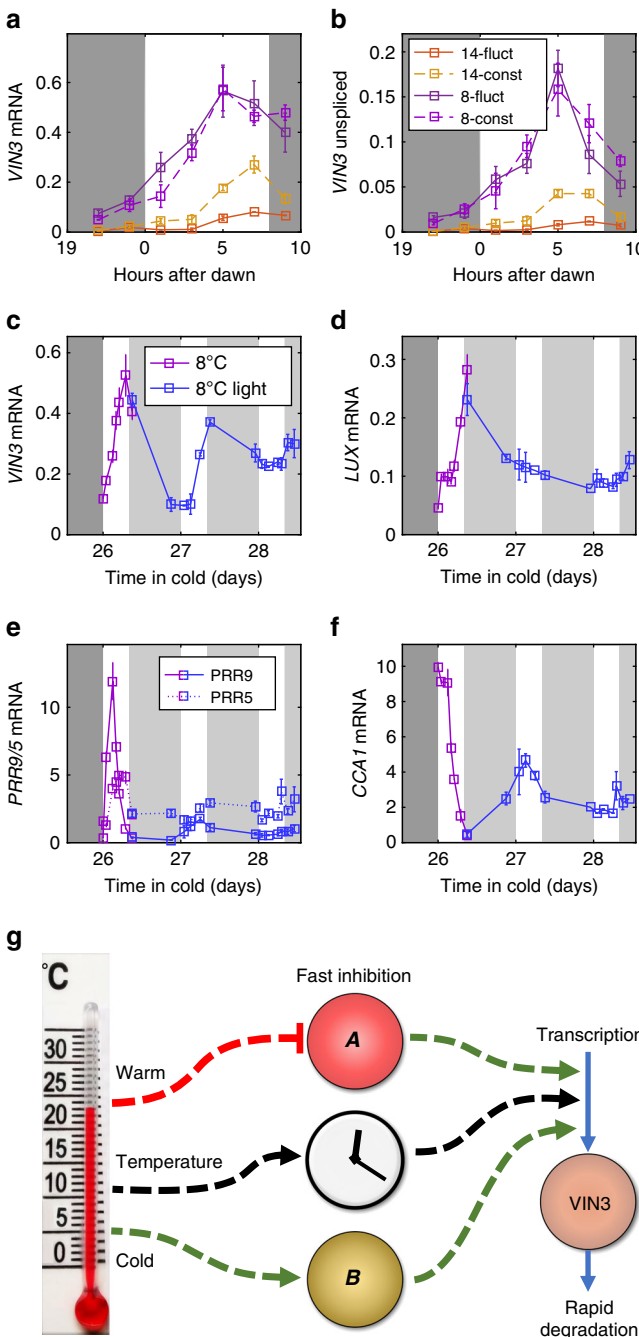

**Fig. 4** *VIN3* expression is regulated in a circadian manner. **a**, **b** *VIN3* spliced and unspliced expression over 12 h. Plants grown in 20 °C night, 22 °C day 16 h photoperiod for 1 week and transferred to vernalising 8 h photoperiod with constant or fluctuating 14.2 or 8 °C profiles (as shown in Fig. 2a) for 4 weeks. The dark grey background indicates night-time. Error bars are s.e.m., $n = 2–3$, average > 2.7. **c–f**, *VIN3* and various clock gene mRNA levels sampled on consecutive days at 8 °C after 4 weeks constant 8 °C, 8 h photoperiod (purple). On the first day of sampling the plants were transferred to constant light conditions (blue). The lighter grey background indicates subjective night. RNA levels normalised to *UBC*, *PP2A*. $n = 1–3$, average > 2.1. **g** Summary of *VIN3* regulation showing the slow regulator ($B$), the new component ($A$) found to be necessary from modelling which gives the absence of warm regulation and the circadian clock, which could itself also be used as a temperature input

reported slow upregulation in response to cold (Fig. 2b, Supplementary Figs. 5 and 6)[4,25–27]. However, every night VIN3 expression returned to very low levels, again indicating that the absolute VIN3 mRNA level does not hold a memory of this slow cold exposure, as predicted by our previous analysis.

Given this diurnal regulation, we next returned to our model to re-evaluate our earlier conclusions. In fluctuating conditions, the daily maximum VIN3 levels increase with time under cold treatment (Supplementary Figs. 5 and 6) confirming the presence of the slow response. The degradation rate in the warm was estimated from plants that were transferred from the cold to the warm in the morning (Fig. 3c, d), when VIN3 levels would otherwise increase as part of its diurnal pattern (Fig. 4a). Therefore, the observed dynamics in Fig. 3c, d will be a composite of the decrease due to the warm and the increase due to the diurnal regulation. Thus, the true degradation rate must be at least as fast as we observe. Hence, as with the model of Fig. 3a, such fast degradation would prevent the slow increase in VIN3 in fluctuating conditions. Therefore, we again conclude that VIN3 expression is regulated by at least two independent temperature inputs, a fast response to warmth and a slow response to cold, and also by a daily rhythm, that is either in addition to, or in combination with, one of the temperature inputs.

To understand whether the diurnal rhythm was a response to light or an influence of the circadian clock, we grew plants in short days and then transferred them to constant light and sampled during the transition. The VIN3 oscillation initially persisted at both mRNA and unspliced levels (Fig. 4c and Supplementary Fig. 7a), unlike for the clock oscillator genes LUX ARRHYTHMO (LUX), PSEUDO-RESPONSE REGULATOR 5 (PRR5), PSEUDO-RESPONSE REGULATOR 9 (PRR9) and others (Fig. 4d, e, Supplementary Fig. 7b, c), which substantially lost cyclic expression within 16 h. However, during the second day in constant light, the VIN3 oscillation was eventually lost, similar to the core oscillator CIRCADIAN CLOCK ASSOCIATED 1 (CCA1) (Fig. 4c, f and Supplementary Fig. 7a). These results are consistent with circadian regulation of transcription[30] and, more specifically, with the presence of 'Evening Element' binding sites in the proximal VIN3 promoter, to which CCA1 has been reported to bind[31]. The clock is also regulated in response to temperature[32–34], suggesting that the circadian clock could form a further temperature input into VIN3 dynamics.

## Discussion

These analyses have shown that, contrary to the current view, multiple and diverse aspects of the temperature profile are monitored to effect vernalisation. Transient cold temperatures (below around 14 °C, based on the vin3−4 mutant, Fig. 2d) during autumn, result in FLC transcriptional downregulation, but not epigenetic silencing. This is likely to involve cold-induced FLC antisense transcripts, whose expression is mutually exclusive with FLC[35]. The subsequent epigenetic silencing of FLC, which is VIN3-dependent, has multiple temperature inputs that operate over different timescales. A schematic of these inputs is shown in Fig. 4g, although more work will be necessary to develop a full model of VIN3 that can be tested experimentally. However, from the existing data we can see that the combined action of these temperature inputs ensures that the major factor regulating VIN3 expression is the absence of spikes of warm temperature (above around 15 °C) each day. In the absence of these spikes (and therefore under continuous cold temperatures), daily peak VIN3 expression is permitted to slowly increase with accumulated time in the cold, in a degree days-like fashion. The exploitation of temperature extremes to generate a signal is reminiscent of a mechanism recently found in the process of seed germination[36],

where temperature fluctuations form a positive signal and induce seed germination. Variation sensitivity may therefore be a general and important mode of information processing in plants.

Separation of slow cold-memory and rapid response (Fig. 4g) also allows priming of VIN3 expression, where previous cold exposure results in faster initial upregulation in response to a second exposure, as observed in both Arabidopsis thaliana and Arabis alpina[26,37]. Previous modelling analyses have incorporated degree day-type dynamics[9,11,14], which in Arabidopsis halleri have been shown to reflect an averaging of 6 weeks of previous temperature exposure[10], but had not identified the requirement for absence of warm. A third input into VIN3 expression is through diurnal variations driven via the circadian clock, possibly via CCA1[31]. All three inputs (Fig. 4g) appear to regulate VIN3 principally through initiation of transcription. Although we do not know the molecular actors themselves, the slow nature of B is consistent with the previously observed accumulation of chromatin marks at VIN3[27]. Furthermore, at least one component of the temperature response does not require new protein synthesis to allow VIN3 upregulation in cold temperatures[38].

Overall, we find that transient cool temperatures ('cold spikes', for example, the cool nights of autumn) are sufficient for the initial, temporary shutdown of FLC transcription. However, the absence of warm temperatures ('warm spikes') is the epigenetically recorded signature of winter for Arabidopsis thaliana (summarised in Supplementary Fig. 8). In the context of a warming and less stable climate, this sensitivity of developmental timing mechanisms to variability in temperature, rather than its average, could have large implications in ecological and agricultural contexts[17,39–41].

## Methods

**Plant growth conditions and field experiments**. The standard vernalisation reference accession Col FRI^SF2 and the mutant vin3–4 FRI have been described previously[38,42]. For field experiments, seeds were stratified at 4 °C for 3 days and sown in 5.7 cm pot trays (Pöppelman, Lohne, Germany), using F2 compost (Levington Horticulture, Suffolk, UK) plus grit (Norwich) or P-soil (Hasselfors, Örebro; Sweden), with six replicate cells per genotype per timepoint for RNA samples. For flowering time assays, seeds were sown in 3.9 cm pot trays (Pöppelman) and thinned to a single plant per cell after germination, with 36 plants per genotype.

In Norwich, trays were placed on benches in a well-ventilated, unheated, unlit glasshouse, and bedded in vermiculite to simulate bedding in soil. Trays were covered with plastic lids or cling film for the first week to ensure germination, and watered when necessary. Plants were sown on 29 September 2014. Seedlings were thinned as necessary until only three per pot were left. Genotypes were randomised[43] within six single-replicate sample-sets per timepoint, and sample-sets were then randomised in a three-block design lengthwise along the greenhouse, with some adjustment to ensure each of the two replicates per block were on different benches. TinyTag Plus 2 dataloggers (Gemini Data Loggers (UK) Ltd) were placed on all benches and in each block at plant level. The lower temperature limit observed of ~2 °C probably resulted from buffering of the lowest temperatures by glasshouse confinement, as loggers placed just outside the glasshouse or very near the vents mirrored internal temperatures closely except below ~4 °C.

For sampling, aerial parts of seedlings (up to 5 weeks) and thereafter the young leaves and meristems were taken from at least three plants per sample, excluding flowering bolts where applicable, and flash frozen in liquid nitrogen. Sampling was performed between 1145–1300 hours.

In Sweden, trays were initially grown outside under plastic covers for 2 weeks at Mid Sweden University, Sundsvall (Sweden North) or Lund University (Sweden South) to ensure germination. Trays were then moved to the experiment sites (North: Ramsta (62° 50.988'N, 18° 11.570'E), South: Ullstorp (56° 06.6721'N, 13° 94.4655'E)) and the trays dug into the soil. Plants were sown and moved on: North Set first planting sown 26 August 2014, moved 11 September; North Set second planting sown 8 September 2014, moved 24 September; South sown 24 September 2014, moved 8 October. Timepoint sample-sets were randomised in three blocks. Trays were watered when necessary. TinyTag Plus 2 dataloggers were placed with the plants at soil level.

For sampling, whole plants were rinsed, dried on paper, placed in sample tubes and frozen on dry ice. Sampling times were between 1100 and 1700 hours and precise times were recorded on each occasion.

Flowering time was scored as the first day of the floral transition being visible at the apical meristem.

**Laboratory experiments**. Plants were sown as for the Norwich glasshouse, but on P24 tray cells (Desch PlantPak, Waalwijk, Netherlands) covered with fine net curtain to allow clean extraction of seedlings. Plants were initially grown at 22 °C 16 h day/20 °C 8 h night for one week ('NV', Non Vernalised), before moving to Panasonic MLR-352 series growth cabinets set to 37–52 μmol light (setting 3) for 8 h per day and the described temperature setting. For the 24 h experiment in Supplementary Fig. 5, an alternative 'NV' temperature regime was used in the Panasonic cabinets as described in the figure. For Fig. 2e, and Supplementary Fig. 3b were returned to 'NV' conditions after vernalisation treatment in the cabinets. For Supplementary Fig. 3b, flowering time plants were also pricked out into individual P24 cells after transfer. For the post-cold degradation experiments of Fig. 3c, d, following the standard NV treatment, plants were moved to 5 °C 8 h photoperiod walk-in vernalisation room for 4 weeks and finally transferred to a growth cabinet set to 22 °C 16 h photoperiod during sampling. For Fig. 4c–f and Supplementary Fig. 7 plants were grown on Murashige and Skoog (MS) agar plates without glucose. They were grown at 20 °C 8 h photoperiod in a growth cabinet for 4 weeks and then moved to 8 °C 8 h photoperiod for 26 days before sampling. On the 26th night, they were separated so that the '8 °C Light' samples were treated with constant light.

**Sample sizes and replication**. For field experiments, six replicates per genotype were chosen to allow for losses in the field and for duplication within randomisation blocks. For flowering time, within space constraints enough individual plants were sown to ensure at least ten samples per randomisation block/condition. For cabinet experiments, three biological or technical replicates were sampled per timepoint to allow sufficient number of samples for statistical analysis while allowing for space constraints within cabinets. Separate sowing dates were used for three replicates of experiments for Figs. 2b–d, 3b, e and 4a, b and Supplementary Figs. 5 and 6, with multiple timepoints sampled. For Fig. 2e, separate sowing dates were used for two replicates of the experiment and two technical replicates were sampled within each experiment. For Figs. 3c, d and 4c–f, and Supplementary Figs. 7 and 9 three technical replicates were taken from the same experiment. Where resulting samples are smaller, this is due to experimental or processing loss (e.g., death of plants in the field, or degradation due to poor sample quality or processing, see 'RNA preparation and QPCR'). Final sample sizes for all data points can be found in Supplementary Data 1.

**RNA preparation and QPCR**. RNA was extracted as described[44], using phenol equilibrated to pH8, followed by lithium chloride precipitation. RNA was DNase treated with Turbo DNA Free DNase, lithium precipitated, then treated with SuperScript Reverse Transcriptase III (both Life Technologies) and gene-specific primers (VIN3 qPCR 1R, FLC spliced R, UBC_qPCR_R, PP2A R2, FLC unspliced_RP 5′-CTTTGTAATCAAAGGTGGAGAGC-3′, VIN3 unspliced R, in one reaction, UBC_qPCR_R, PP2A R2, LUX R, GI R, JF119-CCA1-R, JF117-TOC1-R, JF310-PRR9-R, JF283-PRR5-R in another) to synthesise cDNA. QPCR was performed using SYBRGreen Master Mix II on a LightCycler 480 II (both Roche) using primer pairs: VIN3 qPCR 1F 5′-TGCTTGTGGATCGTCTTGTCA-3′ and VIN3 qPCR 1R 5′-TTCTCCAGCATCCGAGCAAG-3′, FLC spliced F 5′-AGC-CAAGAAGACCGAACTCA-3′ and FLC spliced R 5′-TTTGTCCAGCAGGTGA-CATC-3′, VIN3 unspliced F 5′-GGTTTTATTGCGCGTATTGC-3′ and VIN3 unspliced R 5′-CCAGCATCTGTAAGCACTCG-3′, UBC qPCR F 5′-CTGCGACTCAGGGAATCTTCTAA-3′ and UBC qPCR R 5′-TTGTGCCATT-GAATTGAACCC-3′, PP2A F2 5′-ACTGCATCTAAAGACAGAGTTCC-3′ and PP2A R2 5′-CCAAGCATGGCCGTATCATGT-3′, LUX F 5′-AGGTGGAAGCG-CAAATGAGA-3′ and LUX R 5′-TGTAGACACCAAGAACCTATCTCT-3′, and GI F 5′-GTATCTGCAACGCCAGCGA-3′ and GI R 5′-GCAACTCCCTTT-CAGCCTGA-3′[30], JF118-CCA1-F 5′-CTGTGTCTGACGAGGGTCGAA-3′ and JF119-CCA1-R 5′-ATATGTAAAACTTTGCGGCAATACCT-3′, JF116-TOC1-F 5′-ATCTTCGCAGAATCCCTGTGATA-3′ and JF117-TOC1-R 5′-GCACC-TAGCTTCAAGCACTTTACA-3′, JF309-PRR9-F 5′-GATTGGTGGAATTGA-CAAGC-3′ and JF310-PRR9-R 5′-TCCTCAAATCTTGAGAAGGC-3′, JF282-PRR5-F 5′-GTGTATGTTGAAAGGTGCGG-3′ and JF283-PRR5-R 5′-AGGAG-CAAGTGAAGTTTGTC-3′[45].

Test amplicons were normalised to the geometric mean of two standard reference genes, At1g13320 ('PP2A') and At5g25760 ('UBC')[46,47]. Samples with very low values of control genes (indicating degradation, inhibition or sample loss) and data for amplicons that were below the detection limit were excluded. The final sample size for all experiments can be found in Supplementary Data 1. For the field experiments and for the experiment shown in Supplementary Fig. 5, the output was analysed using LinReg[48] and in order to normalise across a large number of samples, samples were further normalised to a Control Sample run on every QPCR plate. The Control cDNA was made in the same way as the test samples from an equal mix of RNA from 8 weeks cold-treated Col FRI^SF2 and from non-vernalised Col FRI^SF2 to ensure expression of all amplicons and avoid divide-by-zero error, mixed, prediluted and aliquoted to ensure consistency. For other expression experiments, smaller sample numbers allowed simultaneous processing, so Ct values from the LightCycler software were analysed using the ΔΔCt method.

**Mathematical model**. The model was solved numerically[49] with initial conditions (indicated by the subscript 0): $v_0 = 0$ and $r_0 = 0$ for Fig. 3e and $v_0 = 0.124$ and $r_0 = d_1 v_0/p_1$ for Fig. 3d. Due to the simplicity of the model, parameters could be fitted to the data of Fig. 3d, e (constant temperature only) by least squares fitting for $p_1$ and manual adjustment for other parameters: $p_1 = 0.91$ day$^{-1}$, $d_1 = 100$ day$^{-1}$, $d_2 = 7$ day$^{-1}$. Fig 3d provides lower bounds for $d_1$ and $d_2$. The conclusion of the model is not sensitive to the parameters except the decay rate of $r$ in the warm ($d_2$) which is constrained by the data. The temperature dependence of $d_r(T)$ is chosen as a step function at 22 °C so as to use the simplest and most conservative form. Using a temperature step higher than at 22 °C would not replicate the fast degradation observed in Fig. 3c, d, while a step at a lower temperature would lead to similar results in 14-fluctuating conditions but with even lower VIN3 levels. A more complex temperature dependence for the production/degradation of $r$ is possible and indeed necessary to capture the full degree days-like accumulation of VIN3 at different constant cold temperatures. However, even with such additional complexity, the fluctuating temperature behaviour of VIN3 is not captured and therefore the conclusion of the need for two independent thermosensory pathways is not affected. An additional model considering unspliced VIN3 as an intermediate was also tested and the conclusions of the model were the same (Supplementary Fig. 9).

**Statistics**. One-tailed t-test was performed to test for reactivation after the cold (Fig. 2e) (H$_0$: The true mean difference $FLC_{T7} - FLC_{T0}$ is less than or equal to 0). Samples from the same experimental replicate that had experienced the same length of vernalisation treatment were paired. Samples refer to the mean of any technical replicates within the experimental replicate. The same test was performed for the 14-constant treatment (not significant, $p = 0.7480$) and the 14-fluctuating treatment (significant, $p = 0.0064$), using the Matlab function ttest().

To compare the rate of decrease of FLC in the field (Fig. 1 and Results section 'FLC silencing can be separated into two phases in the field'), ANCOVA was performed using the R function lm()[50].

**Code availability**. The authors declare that all code supporting the findings of this study is available from the corresponding author upon request.

**Data availability**. The authors declare that all data supporting the findings of this study are included in the main manuscript file or Supplementary Information or are available from the corresponding author upon request.

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

## Acknowledgements

The authors would like to thank the entire Dean group and friends for help with the field experiments; Huamei Wang and Annie McGee for seed preparation; family Öhman (Ramsta) and Nils Jönsson (Ullstorp) for hosting the field experiments and experimentalists in Sweden; Catherine Taylor and JIC Horticultural Services for plant care in Norwich; Dr. Jie Song (Imperial College London) for the *VIN3* unspliced primers; the Dean group, Dr. Giuseppe Facchetti, Dr. Cecilia Lövkvist, Dr. Laura Dixon, Dr. Xiaoqi Feng and Dr. Steven Penfield (John Innes Centre), Dr. Melissa Antoniou-Kourounioti (University of East Anglia), Dr. Akiko Satake (Kyushu University) and Dr. Karissa Sanbonmatsu (Los Alamos National Laboratories) for discussion; and Ingalill Thorsell of Drakamöllan Gårdshotell for inspirational cooking. This work was funded by the European Research Council grant 'MEXTIM', and supported by the BBSRC Institute Strategic Programmes GRO (BB/J004588/1) and GEN (BB/P013511/1).

## Author contributions

J.H. and R.L.A.-K. designed experiments, acquired data and analysed data for all figures. R.L.A.-K. and M.H. performed mathematical modelling. C.S., K.B., S.H. and T.S. designed experiments, acquired data and analysed data, J.I. designed experiments, D.C. and B.R.C.H. designed experiments and acquired data for Fig. 1. R.H.B. designed experiments, acquired data and analysed data for Figs. 2–4. C.D. and M.H. are responsible for the concept of the research, designed experiments and analysed data. R.L. A.-K., J.H., M.H. and C.D. wrote the paper.

## Additional information

**Competing interests:** The authors declare no competing financial interests.

