## [Peer Review File · Nature Communications]

Reviewers' comments:

Reviewer #1 (Remarks to the Author):

The manuscript by Hepworth et al compared FLC and VIN3 expression profiles in *Arabidopsis thaliana* in field experiments in very different winter temperature conditions to understand the process of vernalization in the wild. In the warmest temperature condition (Norwich), FLC decreased without activation of VIN3 at the initial stage of cold exposure. To explore the difference between VIN3-independent and -dependent vernalization processes, authors performed temperature manipulation experiments in the laboratory, and demonstrated that high temperature spikes influences the VIN3-dependent process by preventing high levels of VIN3 expression, while the effect of high temperature seems to be less pronounced in the VIN3-independent process. A mathematical model that considers a hypothetical variable that regulates VIN3 were developed to explain VIN3 expression profiles in the laboratory. The discrepancy between model prediction and data in Fig. 3e was not resolved, and authors suggested more complicated hypothesis that the circadian clock could be a temperature input into VIN3 dynamics.

Monitoring expression profiles of key genes in the vernalization pathway in the wild is novel and development of mathematical models to explain expression profiles in the wild is important to understand how natural plants respond to fluctuating temperature conditions. However, unfortunately, this manuscript was not successful to advance our understanding because authors failed to explain/interpret their data. There are five major concerns.

- (1) Although there are significant differences in FLC and VIN3 expression profiles between three different field sites under different temperature regimes, mechanisms underlying the differential expression profiles never been explained. In North Sweden, VIN3 was activated around 30 days after sowing, but FLC expression levels decreased continuously throughout the monitoring period. This result seems to be inconsistent with the argument that VIN3-independent and VIN3-dependent phases can be separated clearly. It is better to use the same scale for x axis in Fig. 1c.
- (2) Authors failed to develop mathematical models that can be used to predict expression profiles of FLC and VIN3 in the wild. The model used to explain VIN3 dynamics in the laboratory condition is too simplistic to apply to natural plants. To demonstrate the importance of high temperature spike in vernalization in the wild, mechanistic explanation about the relationship between temperature and expression profiles are necessary. It seems strange that VIN3 was elevated in the fall in the North Sweden site even when high temperature occurred.
- (3) Authors presented the data for diurnal cycle of VIN3 and suggested that the circadian clock could be a temperature input into VIN3 dynamics. However, the model presented in Fig. 3 did not consider the effect of circadian clock. It is strange that authors used the model neglecting diurnal dynamics even when they stressed the importance of circadian clock.
- (4) A mathematical model considers a hypothetical variable r that regulates VIN3. However, there were no arguments about the potential candidate for this variable.
- (5) How differential expression profiles in FLC and VIN3 influence flowering time was not explained. Lack of data on phenotypic response reduces the impact of the manuscript.

Minor comments

- (1) Fig. 1b: There seems to be a lower bound for temperature in Norwich. Any trouble for temperature sensor?
- (2) Fig. 3: It's better to plot the profiles of r under each temperature condition.
- (3) Fig. 4: What is A? It was not included in the model presented in Fig. 3.

Reviewer #2 (Remarks to the Author):

In this manuscript the authors investigate the repression of FLC in plants from the common laboratory *Arabidopsis thaliana* strain, ColFRIsf2, grown in the field at three difficult locations. This study demonstrated that vernalization can be divided into two phases, one that is VIN3-independent and second which is dependent on VIN3. In a second part to this study the authors investigated the regulation of VIN3 in response to different temperatures in controlled environment cabinets, which has allowed them to dissect VIN3 transcription under constant or fluctuating temperatures, more similar to those experienced in the field. They develop a mathematical model for VIN3 regulation; this reveals that there are two components to VIN3 regulation – one that regulates transcription at low temperatures and a second, which they have termed an “absence of warmth factor” that shuts down VIN3 transcription at high temperatures.

It is already well recognized that cold snaps in autumn do not lead to an effective vernalization response so the observations per se are not really novel, but what has been lacking is an understanding of just why this should be. This study provides some insights into what other environmental signals play a role in regulating vernalization, at least for *Arabidopsis*, as it seems that the “absence of warmth” factor is the key for discriminating between autumn and winter. The concept that plants are using multiple inputs to regulate vernalization, adds to our understanding of this well characterized response. It will be interesting to see whether a similar model can be applied to cereals where the regulation of vernalization is quite different to that in the Brassicaceae.

I do have some criticisms on the manuscript, but I should say at on outset that while I can assess the biology, I am not able to evaluate the mathematical model presented here.

1. The observation that there is a VIN3 independent repression of FLC is not novel. It has been demonstrated both by this group (Swiezewski et al., (2009) *Nature* 462: 799-) and others (Helliwell et al., 2011; *Plos ONE* 6:e21513) that FLC is repressed in a *vin3-4* mutant.
2. Lines 76-83; The authors claim that the “changed VIN3 behaviour in Norwich as compared to the two Swedish sites could be the cause of the disparity in FLC dynamics” I did not find this very satisfying as it is far from clear why the rate of FLC repression should be greater in Norwich than in Sweden after VIN3 induction nor why the dynamics of FLC repression should differ at the two Swedish sites.
3. Lines 127-135; While the mechanism leading to FLC repression has not been elucidated, it has been proposed that the initiation of repression may result from changes in FLC chromatin at low temperatures (Helliwell et al (2015) *Trends Plant Sci* 20: 76); it was also proposed that the rate of repression would vary with temperature, which would be consistent with the greater efficiency of VIN3 independent repression at fluctuating 14^o C where the temperature drops as low as 8^o C, than at constant 14^o C. Here the authors only present data for FLC mRNA (which equals the difference between the rate of FLC mRNA production/FLC transcription – mRNA degradation), so one can't determine whether the differing repression rate seen in the different environments (Norwich vs Sweden) or temperature regimes is due to a different transcription rate or degradation rate. I suspect the former but looking at FLC unspliced mRNA might give some clues.
4. Figure 2(e) and associated text; it is not surprising that the VIN3 independent repression of FLC is at least partially stable because it has already been shown that absence of FLC transcription is sufficient to allow accumulation of H3K27me3 in the absence of any vernalization treatment and thus in the absence of VIN3 activity (ref # 18 Buzas et al., (2011)).
5. Line 148-149; I suggest replacing ref #24: Bond et al., (2009) *Plant J* with Bond et al (2009) *Molecular Plant* 2: 724- 737. Reference 24 concerns VIN3 induction in response to low oxygen stress which differs from that seen under low temperature conditions (reported in the *Molecular Plant* paper). The *Molecular Plant* paper also shows that there is no need for protein synthesis for VIN3 to occur induction at low temperatures, making it unlikely that the model component “r” is a newly synthesized

protein.

6. Line 158-161; These observations could be explained by data from Bond et al Molecular Plant, which shows that there is priming effect at VIN3 chromatin such that if the temperature increases and VIN3 transcription ceases, then re-exposure to low temperatures allows more rapid induction than would normally occur in a plant that had not previously been exposed to low temperatures. This priming effect would facilitate the gradual increased in VIN3 transcription under the fluctuating 14^o C regime reported here, as it would predict that the transcription rate would not be reset to zero each day.

7. I found the terminology "the absence of warmth component" counter-intuitive as that component is present when the temperatures increase about the threshold. I wonder if the authors would consider changing this to the "warmth component". Absence of the warmth component would therefore ensure that VIN3 transcription continued to increase.

8. Figure 3; It is not clear to me where the data for panels (c) and (d) comes from – which temperature regime??. I also query the scale on the Y axes in these same two panels. If I understand this correctly it implies that VIN3 transcripts drop below the starting point in NV plants a few hours after the temperature increases??

9. Line 185-186; it might be more productive to look at FLC unspliced message to monitor any rapid response as it has previously been shown that the rate of degradation of FLCmRNA is very slow (Csorba et al., 2014 PNAS 111: 16160-).

10. I appreciated that the authors have indicated the extent of replication, particularly where they do not really have adequate numbers of samples. In general there is sufficient data over the time courses so that the loss of a few samples is not a cause for concern.

Reviewer #1 (Remarks to the Author):

The manuscript by Hepworth et al compared *FLC* and *VIN3* expression profiles in *Arabidopsis thaliana* in field experiments in very different winter temperature conditions to understand the process of vernalization in the wild. In the warmest temperature condition (Norwich), *FLC* decreased without activation of *VIN3* at the initial stage of cold exposure. To explore the difference between *VIN3*-independent and *VIN3*-dependent vernalization processes, authors performed temperature manipulation experiments in the laboratory, and demonstrated that high temperature spikes influences the *VIN3*-dependent process by preventing high levels of *VIN3* expression, while the effect of high temperature seems to be less pronounced in the *VIN3*-independent process. A mathematical model that considers a hypothetical variable that regulates *VIN3* were developed to explain *VIN3* expression profiles in the laboratory. The discrepancy between model prediction and data in Fig. 3e was not resolved, and authors suggested more complicated hypothesis that the circadian clock could be a temperature input into *VIN3* dynamics.

Monitoring expression profiles of key genes in the vernalization pathway in the wild is novel and development of mathematical models to explain expression profiles in the wild is important to understand how natural plants respond to fluctuating temperature conditions. However, unfortunately, this manuscript was not successful to advance our understanding because authors failed to explain/interpret their data. There are five major concerns.

(1) Although there are significant differences in *FLC* and *VIN3* expression profiles between three different field sites under different temperature regimes, mechanisms underlying the differential expression profiles never been explained. In North Sweden, *VIN3* was activated around 30 days after sowing, but *FLC* expression levels decreased continuously throughout the monitoring period. This result seems to be inconsistent with the argument that *VIN3*-independent and *VIN3*-dependent phases can be separated clearly. It is better to use the same scale for x axis in Fig. 1c.

Thank you for this comment. The differential expression profiles are, we believe, explained by the relative action of the temperature-dependent *VIN3*-independent and *VIN3*-dependent pathways, as we have now tried to make clearer in the manuscript (lines 75-92). Their action can be separated in some temperature profiles, but only when the presence of warm spikes occurs during the day to suppress the *VIN3*-dependent phase. In North Sweden, the duration of the presence of warm spikes (a little over 30 days) is too short for us to be able to estimate a slope for the early *FLC* phase in this case. However, the *VIN3* levels are indeed exactly as we would expect given our absence of warmth criterion. They are much lower for the first 3 sampling times, while there are still peaks to high temperature. We have added Extended Data Fig. 2 which shows in greater detail this result, and made the point in the main text (line 134). Following the early increase in *VIN3* levels, we would expect a single behaviour in *FLC* downregulation, given by the combination of the two pathways, which is indeed what we see and is also the case in South Sweden (though with a different overall rate of downregulation in South versus North Sweden).

We have adjusted Fig. 1c to show the same x axis scale. Due to the different duration of the experiments and to avoid empty space, the size of the figures was adjusted. Furthermore, Fig. 1d allows direct comparison of the different sites.

(2) Authors failed to develop mathematical models that can be used to predict expression profiles of *FLC* and *VIN3* in the wild. The model used to explain *VIN3* dynamics in the laboratory condition is too simplistic to apply to natural plants. To demonstrate the importance of high temperature spike in vernalization in the wild, mechanistic explanation about the relationship between temperature and expression profiles are necessary.

The model shown in the paper is only intended to test the simplest hypothesis to ensure that we are not introducing unnecessary complexity. We have therefore rewritten the manuscript to make it clear that the model is explicitly rejected because it cannot reproduce the simple laboratory experiments, and therefore more than one temperature regulator is required. This is why we do not attempt to make any field predictions from this model. This rewriting includes renaming the regulators that we think do exist as 'A' and 'B', so that we can reject '*r*' more clearly. We have indeed developed a more detailed model that captures the behaviour of *VIN3* and *FLC* in the lab and in the field, but this model is much more complex. We are therefore preparing a separate, much longer manuscript to describe the detailed model. Combining both the results of the current paper and those of the detailed model would make for a manuscript with simply too much information and analysis for the conclusions to be easily absorbable. We have therefore chosen to split the results into two.

It seems strange that *VIN3* was elevated in the fall in the North Sweden site even when high temperature occurred.

We have now added the new Extended Data Fig. 2 as mentioned previously, which shows that in North Sweden the *VIN3* levels were elevated exactly when high temperatures ceased, as expected.

(3) Authors presented the data for diurnal cycle of *VIN3* and suggested that the circadian clock could be a temperature input into *VIN3* dynamics. However, the model presented in Fig. 3 did not consider the effect of circadian clock. It is strange that authors used the model neglecting diurnal dynamics even when they stressed the importance of circadian clock.

The simple model of Fig. 3 failed to adequately capture two key features of the laboratory data. Specifically, the model was unable to reproduce both the slow increase and fast decrease of *VIN3* transcription in response to different temperatures. Adding circadian behaviour would make the simple model significantly more complicated. However, as the reviewer suggests, it is important to point out why we neglected the diurnal dynamics in our modelling and how including it would affect our results. The diurnal pattern also cannot explain the observed behaviour, and we have added text to the results section of the manuscript which we hope makes that clear (lines 216-227).

(4) A mathematical model considers a hypothetical variable *r* that regulates *VIN3*. However, there were no arguments about the potential candidate for this variable.

We initially made no suggestions as to candidates for *r* as we did not want to make any assumptions which might influence our analysis. However, we did mention that *cis*-chromatin changes previously identified at the *VIN3* locus would be consistent with *r* (line 163-4).

Although we then reject this version of *r*, we then split its properties into two thermosensors, which for clarity in the revised manuscript we describe as A and B. As the reviewer suggests we now discuss these further (lines 270-274). Indeed, Reviewer 2 also helpfully suggests (5) a further point of evidence on the nature of one of these thermosensors, which we have included. However, it is beyond the scope of this work to do the extensive mutant screening required to identify the necessary factors, as there are, to our knowledge, no molecular components published that act as thermosensors over this temperature range.

(5) How differential expression profiles in *FLC* and *VIN3* influence flowering time was not explained. Lack of data on phenotypic response reduces the impact of the manuscript.

We have added flowering time data, which shows that the *vin3-4* mutant flowers a month later than the reference line in the field in Norwich (Extended Data Fig. 3a). This is also supported by the results in Wilczek *et al.* 2009 (reference 9). We have also included data from the fluctuating and

constant conditions experiments (Extended Data Figure 3b) which show that the *vin3-4* mutant flowers later than the wild type, but is nevertheless promoted by the VIN3-independent repression of *FLC*. Additionally, there is a very wide body of evidence, much of which has been cited in the paper, that has long established that varying *FLC* levels cause quantitative changes in flowering time.

(1) Fig. 1b: There seems to be a lower bound for temperature in Norwich. Any trouble for temperature sensor?

Multiple temperature sensors were used to record the temperature inside the glasshouse in Norwich, so we are confident this is not a sensor problem. A possible explanation (glasshouse confinement buffering lower temperatures) is given in the Methods (lines 297-300). We have also now added this explanation to the Fig. 1 legend, as this is likely to be a common point of confusion.

(2) Fig. 3: It's better to plot the profiles of r under each temperature condition.

A figure has been added to the Extended Data (Extended Fig. 4) showing the profiles of r in comparison to those of *VIN3*. This is mentioned in the main text (lines 170-172) explaining why the profiles are practically identical.

(3) Fig. 4: What is **A**? It was not included in the model presented in Fig. 3.

Indeed, we do not include **A** in the model of Fig. 3. In that first model, we try to explain our observations with r only and find that it is not possible. This leads us to conclude that **A** is necessary. To make this point clearer, the legend of Fig. 4 has been amended.

Reviewer #2 (Remarks to the Author):

In this manuscript the authors investigate the repression of *FLC* in plants from the common laboratory *Arabidopsis thaliana* strain, ColFRIsf2, grown in the field at three difficult locations. This study demonstrated that vernalization can be divided into two phases, one that is VIN3-independent and second which is dependent on VIN3. In a second part to this study the authors investigated the regulation of VIN3 in response to different temperatures in controlled environment cabinets, which has allowed them to dissect VIN3 transcription under constant or fluctuating temperatures, more similar to those experienced in the field. They develop a mathematical model for VIN3 regulation; this reveals that there are two components to VIN3 regulation – one that regulates transcription at low temperatures and a second, which they have termed an “absence of warmth factor” that shuts down VIN3 transcription at high temperatures.

It is already well recognized that cold snaps in autumn do not lead to an effective vernalization response so the observations per se are not really novel, but what has been lacking is an understanding of just why this should be. This study provides some insights into what other environmental signals play a role in regulating vernalization, at least for *Arabidopsis*, as it seems that the “absence of warmth” factor is the key for discriminating between autumn and winter. The concept that plants are using multiple inputs to regulate vernalization, adds to our understanding of this well characterized response. It will be interesting to see whether a similar model can be applied to cereals where the regulation of vernalization is quite different to that in the Brassicaceae.

I do have some criticisms on the manuscript, but I should say at on outset that while I can assess the biology, I am not able to evaluate the mathematical model presented here.

1. The observation that there is a VIN3 independent repression of FLC is not novel. It has been demonstrated both by this group (Swiezewski et al., (2009) Nature 462: 799-) and others (Helliwell et al., 2011; Plos ONE 6:e21513) that FLC is repressed in a *vin3-4* mutant.

We have adapted the text and added appropriate referencing to support the point that it is the temperature-response, not the existence, of the VIN3-independent pathway that we present as novel (line 93 to 99).

2. Lines 76-83; The authors claim that the “changed VIN3 behaviour in Norwich as compared to the two Swedish sites could be the cause of the disparity in FLC dynamics” I did not find this very satisfying as it is far from clear why the rate of FLC repression should be greater in Norwich than in Sweden after VIN3 induction nor why the dynamics of FLC repression should differ at the two Swedish sites.

We thank the reviewer for this excellent point. We only tried to explain the “disparity” in terms of the overall shape (two slopes in Norwich, one at each site in Sweden) rather than the difference in the rate of shutdown. The word dynamics is misleading and has been changed in the text (line 85). We believe that the difference in the slopes is due to the low temperatures which are below the optimum for vernalization. The text has been amended to explain this and references have been added (lines 81-83).

3. Lines 127-135; While the mechanism leading to FLC repression has not been elucidated, it has been proposed that the initiation of repression may result from changes in FLC chromatin at low temperatures (Helliwell et al (2015) Trends Plant Sci 20: 76); it was also proposed that the rate of repression would vary with temperature, which would be consistent with the greater efficiency of VIN3 independent repression at fluctuating 14° C where the temperature drops as low as 8° C, than at constant 14° C. Here the authors only present data for FLC mRNA (which equals the difference between the rate of FLC mRNA production/FLC transcription – mRNA degradation), so one can't determine whether the differing repression rate seen in the different environments (Norwich vs Sweden) or temperature regimes is due to a different transcription rate or degradation rate. I suspect the former but looking at FLC unspliced mRNA might give some clues.

Again, this is another important point – as the reviewer suggests, we have added data for unspliced *FLC* (Extended Figs. 2, 5 and 6, plus lines 79-81). This supports the reviewer's suggestion and our assumption that a change in transcription rate is the cause of the change in *FLC* mRNA abundance.

4. Figure 2(e) and associated text; it is not surprising that the VIN3 independent repression of FLC is at least partially stable because it has already been shown that absence of FLC transcription is sufficient to allow accumulation of H3K27me3 in the absence of any vernalization treatment and thus in the absence of VIN3 activity (ref # 18 Buzas et al., (2011)).

Thanks to the reviewer for making this point. We agree with this conclusion, and have added a comment on this point (lines 150-153).

5. Line 148-149; I suggest replacing ref #24: Bond et al., (2009) Plant J with Bond et al (2009) Molecular Plant 2: 724- 737. Reference 24 concerns VIN3 induction in response to low oxygen stress which differs from that seen under low temperature conditions (reported in the Molecular Plant paper). The Molecular Plant paper also shows that there is no need for protein synthesis for VIN3 to occur induction at low temperatures, making it unlikely that the model component “r” is a newly synthesized protein.

The reviewer is quite right, we confused our references here. We have also added a few lines on the potential nature of the molecular actors, and included a reference to the cycloheximide experiment

in the Plant Journal paper as part of this (lines 271-274). However, we cannot currently specifically assign that property to either the 'A' or 'B' responses.

6. Line 158-161; These observations could be explained by data from Bond et al Molecular Plant, which shows that there is priming effect at VIN3 chromatin such that if the temperature increases and VIN3 transcription ceases, then re-exposure to low temperatures allows more rapid induction than would normally occur in a plant that had not previously been exposed to low temperatures. This priming effect would facilitate the gradual increased in VIN3 transcription under the fluctuating 14° C regime reported here, as it would predict that the transcription rate would not be reset to zero each day.

Indeed, we think a possible form of **(B)** is something like this suggestion but we do not have the explicit evidence to support this hypothesis at the moment. However, we do cite the paper mentioned (lines 162-164, 271-274).

7. I found the terminology “the absence of warmth component” counter-intuitive as that component is present when the temperatures increase about the threshold. I wonder if the authors would consider changing this to the “warmth component”. Absence of the warmth component would therefore ensure that VIN3 transcription continued to increase.

The word “component” is perhaps the most misleading part. As we do not know how this thermosensor functions, we cannot say whether it is a repressor present when the temperature is high or an activator present when the temperature is low. The wording has been changed (line 184) to make it clear that we mean a pathway that takes the absence of warm temperatures as a positive signal to upregulate *VIN3*, rather than a single component.

8. Figure 3; It is not clear to me where the data for panels (c) and (d) comes from – which temperature regime??. I also query the scale on the Y axes in these same two panels. If I understand this correctly it implies that VIN3 transcripts drop below the starting point in NV plants a few hours after the temperature increases??

The starting point shown in Fig. 3c,d is after 4 weeks of vernalization at 5°C (not NV), and thereafter data was collected at 22°C. To make this protocol clearer we have added additional text to the legend. The temperature regime is described in the legend and methods, but was not clear in the main text where the figure is introduced. This has been amended to point out that the data shown follows a cold treatment (lines 169).

9. Line 185-186; it might be more productive to look at FLC unspliced message to monitor any rapid response as it has previously been shown that the rate of degradation of FLCmRNA is very slow (Csorba et al., 2014 PNAS 111: 16160-).

We have added data for unspliced *FLC* which also shows no consistent rapid response to temperature (Extended Data Figs. 5 and 6).

10. I appreciated that the authors have indicated the extent of replication, particularly where they do not really have adequate numbers of samples. In general there is sufficient data over the time courses so that the loss of a few samples is not a cause for concern.

We appreciate this remark, thank you.

REVIEWERS' COMMENTS:

Reviewer #2 (Remarks to the Author):

The authors have satisfactorily addressed most of my criticisms in the revision of the manuscript. There are some minor points that I noted in this version that need to be corrected. I've noted these below:

1. P5 line 102; do you really mean 14.2°C or 14°C?
2. P6 lines 146-153; I think that this paragraph needs some rewording. For example the second sentence states: "We observed this response in plants treated at constant 14°C" – but the preceding sentence gave two responses so it would be better to state which of these responses was observed. The final sentence "Indeed, in the *vin3-4* mutant, fluctuating 14°C..." doesn't really follow from the previous sentence about epigenetic silencing of FLC that occurs in the absence of transcription. It would be more logical to link this observation (ie the data on FLC mRNA and flowering time in *vin3-4*) with the relative levels of FLC mRNA at constant and fluctuating 14°C in wt plants where it does not correlate with VIN3 expression.
3. A question that I should have asked in my review of the original version is: at what temperature does the COOLAIR antisense transcript become elevated? I don't believe these data have been published anywhere.

Reviewer #2 (Remarks to the Author):

The authors have satisfactorily addressed most of my criticisms in the revision of the manuscript. There are some minor points that I noted in this version that need to be corrected. I've noted these below:

1. P5 line 102; do you really mean 14.2°C or 14°C?

For simplicity we had only written 14.2°C at the first mention of the conditions. That is the correct value. To avoid confusion, the temperature conditions have now been replaced with "14-constant" and "14-fluctuating" while any mention of the temperature has been changed to 14.2°C.

2. P6 lines 146-153; I think that this paragraph needs some rewording. For example the second sentence states: "We observed this response in plants treated at constant 14°C" – but the preceding sentence gave two responses so it would be better to state which of these responses was observed. The final sentence "Indeed, in the *vin3-4* mutant, fluctuating 14°C..." doesn't really follow from the previous sentence about epigenetic silencing of FLC that occurs in the absence of transcription. It would be more logical to link this observation (ie the data on FLC mRNA and flowering time in *vin3-4*) with the relative levels of FLC mRNA at constant and fluctuating 14°C in wt plants where it does not correlate with VIN3 expression.

The two responses have been replaced with "remain silenced" to capture the observed response (which is "not increasing"). The sentence was moved to the previous paragraph as suggested where indeed it fits much better.

3. A question that I should have asked in my review of the original version is: at what temperature does the COOLAIR antisense transcript become elevated? I don't believe these data have been published anywhere.

We observe COOLAIR induction at 14°C and lower, but we have not tested higher temperatures.